# Treatment Effect of Platelet Gel on Reconstructing Bone Defects and Nonunions: A Review of In Vivo Human Studies

**DOI:** 10.3390/ijms231911377

**Published:** 2022-09-27

**Authors:** Che-Yu Lin

**Affiliations:** Institute of Applied Mechanics, College of Engineering, National Taiwan University, No. 1, Sec. 4, Roosevelt Road, Taipei 10617, Taiwan; cheyu@ntu.edu.tw

**Keywords:** bone fractures, orthopedics, orthopedics, surgery, platelet-rich plasma, regenerative medicine, cell therapy, biomaterials

## Abstract

In ideal circumstances, a fractured bone can heal properly by itself or with the aid of clinical interventions. However, around 5% to 10% of bone fractures fail to heal properly within the expected time even with the aid of clinical interventions, resulting in nonunions. Platelet gel is a blood-derived biomaterial used in regenerative medicine aiming to promote wound healing and regeneration of damaged tissues. The purpose of this paper is to review relevant articles in an attempt to explore the current consensus on the treatment effect of platelet gel on reconstructing bone defects and nonunions, hoping to provide a valuable reference for clinicians to make treatment decisions in clinical practice. Based on the present review, most of the studies applied the combination of platelet gel and bone graft to reconstruct bone defects and nonunions, and most of the results were positive, suggesting that this treatment strategy could promote successful reconstruction of bone defects and nonunions. Only two studies tried to apply platelet gel alone to reconstruct bone defects and nonunions, therefore a convincing conclusion could not be made yet regarding the treatment effect of platelet gel alone on reconstructing bone defects and nonunions. Only one study applied platelet gel combined with extracorporeal shock wave therapy to reconstruct nonunions, and the results were positive; the hypothetical mechanism of this treatment strategy is reasonable and sound, and more future clinical studies are encouraged to further justify the effectiveness of this promising treatment strategy. In conclusion, the application of platelet gel could be a promising and useful treatment method for reconstructing bone defects and nonunions, and more future clinical studies are encouraged to further investigate the effectiveness of this promising treatment method.

## 1. Introduction

Bone is a specialized tissue rich in cells including osteoblasts, osteoclasts, and osteocytes responsible for remodeling [1]. Because of its remarkable remodeling ability, bone is a type of tissue that has high potential for regeneration (i.e., self-healing or self-repair) [2,3]. In ideal circumstances, such as the healing of a simple bone fracture or a tooth extraction socket, a fractured bone can heal properly by itself or with the aid of standard conservative or surgical clinical interventions [1,3]. However, it has been reported that around 5% to 10% of bone fractures fail to heal properly within the expected time even with the aid of clinical interventions, resulting in delayed union or nonunion [4,5,6,7,8,9,10,11,12,13]. Delayed union or nonunion often occurs in fractured bones with critical-size defects, with infections, without proper mechanical stability, or could occur in patients with bone fractures who have certain diseases (such as diabetes) or risk factors (such as smoking) [14,15]. Indeed, bone regeneration is a complex process that needs numerous basic requirements as well as proper biological and mechanical conditions to accomplish, and the lack of one or more of the requirements and conditions may lead to the occurrence of delayed union or nonunion [16]. To date, despite continuous advances in the treatment strategies of bone fractures, delayed union or nonunion is still not uncommon and remains a challenge in orthopedics [4,17,18,19,20]. Please note that, although two different terms “delayed union” and “nonunion” are often used simultaneously to describe this tough clinical scenario as in the text above, it is believed that in many cases it is difficult to clearly indicate that a healing process is just delayed, or impaired with no possibility of healing [3]. Hence, in the following text, the term “nonunion” will be used solely to describe this clinical scenario.

The U.S. Food and Drug Administration defines a nonunion as a fracture that is at least nine months old and has not shown any signs of healing for three consecutive months, while some authors define a nonunion as a fracture that has no possibility of healing without further intervention according to the treating physician [4,20]. The causes of nonunion are complex and multifactorial and are related to the type of fracture, the severity of fracture, as well as the physical conditions, diseases, and risk factors in the patient [3,4,9,11,21,22]. Nonunions usually involve symptoms including pain, deformity, and functional disability [3,20] and could significantly decrease the health-related quality of life of patients and cause associated socio-economic problems [10]. Hence, bone nonunion remains a major public health issue that causes problems in many aspects, and there is a crucial need to better understand the underlying mechanisms and improve the treatment strategies of nonunions in order to meet the needs of patients and reduce the associated problems.

In current clinical practice, bone grafting including autograft and allograft is still the gold standard treatment method used to reconstruct bone nonunions [4,9,23,24,25,26]. The merits of using autograft include safety, effectiveness, no immune rejection, and no disease transmission since the donors themselves are the recipients [9,27]. However, the use of autograft has several disadvantages such as low availability of autologous bones and the risk of symptoms such as pain, infection, hematoma, and necrosis on the donor site [23,24,26,27,28,29]. In addition, if the defect area is larger than a critical size, the success rate could be compromised [3,4,30]. On the other hand, since allogeneic bones are transplanted from cadavers or living donors, the use of allograft can resolve the problems associated with the use of autograft to a certain extent [27]. Importantly, there is no need to harvest allogeneic bones from the recipients themselves with the use of allograft, eliminating the symptoms on the donor sites of the recipients [27]. Unfortunately, in contrast to the merits of using autograft, using allograft may cause the risks of immune rejection and disease transmission [31]. In addition, insufficient availability of allogeneic bones is also a concern [31]. Since both kinds of bone grafting methods have significant disadvantages, it is important to develop alternative strategies to reconstruct bone nonunions and accelerate bone regeneration.

The drawbacks of current clinical treatment methods for reconstructing bone nonunions motivate researchers to search for alternative strategies in the field of biomaterials and tissue engineering [32,33,34,35,36]. The ultimate goal is to develop biomimetic biomaterials to be served as bone grafts for the regeneration of bones [4,37]. For years, numerous kinds of biomaterials have been developed for applications in bone tissue engineering, including the most widely used kinds based on ceramics, hydrogels, and polymers [9,27]. However, to date, despite substantial efforts over a long period of time, most biomaterials intended to be used in bone tissue engineering are still under fundamental investigation and have not been used on human beings in clinical practice. In the field of biomaterials, it seems that there is still a long distance between research in laboratories and real clinical applications. It has been proven that the growth of new bones using current tissue engineering strategies is extremely difficult [3,38]. One of the possible explanations is that a kind of current biomaterial can only possess some but not all of the properties of natural bones and therefore cannot meet all the requirements for bone regeneration [4,9]. In addition, it is believed that the potential of current biomaterials is still limited to the reconstruction of small bone defects [9]. Hence, in recent years, the focus has been shifted to searching for strategies in regenerative medicine that aim to trigger endogenous healing and regeneration responses [3,39].

Platelet gel is a blood-derived biomaterial used in regenerative medicine aiming to promote wound healing and regeneration of damaged tissues [40,41,42]. Platelet gel, a modification of fibrin gel (or termed fibrin glue), is obtained by mixing platelet-rich plasma with activators such as thrombin and calcium chloride to convert fibrinogen in plasma into a fibrin gel [40,41,42,43]. Different from fluid-like platelet-rich plasma, platelet gel is a gel-like material rich in platelets, as its name suggests (Figure 1), rapidly formed when platelet-rich plasma is mixed with thrombin and calcium chloride (Figure 2). Platelet-rich plasma containing a high concentration of platelets is obtained by centrifugating the autologous whole blood of the patient [44,45,46,47]. Mixing platelet-rich plasma with thrombin and calcium chloride not only leads to the gelatinization of plasma, but also leads to the activation of platelets that releases numerous growth factors from platelets into the environment of the tissue [41,43]. Hence, platelet gel can also be viewed as the activated version of platelet-rich plasma with high concentrations of numerous kinds of growth factors. Platelet gel is like a natural scaffold derived from the autologous whole blood of the patient, and it has numerous advantages and tremendous potentials for applications in regenerative medicine: first, unlike platelet-rich plasma which is fluid-like, platelet gel is gel-like and therefore can densely fill the defect and can be fixed at the site of the defect; second, platelet gel contains numerous types and a large amount of growth factors that are essential for promoting healing and regeneration; third, platelet gel can serve as a natural scaffold, providing an appropriate environment for cells and tissues to grow; fourth, when mixing with the bone graft, platelet gel can enhance the structure of the graft and keep the graft firm, while the mixture of graft and platelet gel can be molded to the desired shape to fill the defect and bridge both ends of the defect [15]. The advantages of platelet gel mentioned above are illustrated in Figure 3 and summarized in Table 1. Since first introduced in the early 1990s, platelet gel has been used in numerous clinical applications in clinical practice, including orthopedic surgery, soft and hard tissue surgery, oral and maxillofacial surgery, plastic surgery, implantology, wound healing, treatment of ulcers, and so on [41,42,48].

The purpose of the present paper is to review relevant articles in an attempt to explore the treatment effect of platelet gel on reconstructing bone defects and nonunions. The information provided in the present review paper not only can be useful for clinicians to make treatment decisions in clinical practice, but also can motivate researchers to further investigate the treatment effect and underlying mechanisms of platelet gel on reconstructing bone defects and nonunions. Please note that, since the author of the present paper believes that the conclusions drawn from in vivo studies involving humans as subjects can most appropriately meet the intention of the present paper for providing information for clinicians to directly apply in clinical practice, the studies reviewed in the present paper are those in which the subjects are humans while the studies involving animals as subjects are not included.

## 2. Search Strategy for the Articles for Reviewing

Studies using platelet gels in the clinical applications relevant to the reconstruction of bone defects in vivo were searched using two electronic databases including PubMed and Google Scholar. The combinations of keywords were used for the search, and the keywords included platelet gel, platelet gels, platelet-rich plasma gel, platelet-rich plasma gels, bone, bones, bony, defect, defects, fracture, fractures, non-union, non-unions, nonunion, nonunions, surgery, and surgeries. One of the four keywords, platelet gel, platelet gels, platelet-rich plasma gel, or platelet-rich plasma gels and one of the remaining keywords described above, must have been included in the title of the article. For each database, the time period for the search was from the earliest available date to 18 August 2022. The inclusion criteria for the search were: (1) original research articles and case studies using platelet gels in the clinical applications relevant to the reconstruction of bone defects in vivo; (2) the subjects involved are humans. The exclusion criteria were: (1) the article was not an original research article or a case study; (2) the study did not belong to the field of orthopedics, and the type of the tissue investigated is not bone; (3) the study was not an in vivo study; (4) the subjects involved are animals; (5) the language of the official website of the article is not English.

There were 15 articles that met the search criteria. In the following, the main findings and methods of these articles are concisely and comprehensively described. The presentation below is organized into three sub-sections, categorized on the basis of the types of medical problems for which platelet gels are used in treatment. In each sub-section, the review of each article is presented in chronological order of publication. The treatment effect (positive or others) is mentioned in the first sentence in each paragraph describing the review of each article. The treatment strategy and effect of each article are summarized in Table 2.

## 3. Research on Using Platelet Gel to Reconstruct Bone Nonunions

In a study presented by Rughetti et al. [49], the authors showed a positive treatment effect of using autologous platelet gel combined with extracorporeal shock wave therapy on reconstructing nonunions. Fifteen patients suffering from nonunions at long bones (femur, tibia, humerus, radius, or ulna) were enrolled. They all had an old fracture, and it was diagnosed as a nonunion since it had been treated using immobilization and extracorporeal shock wave therapy but no callus formation could be observed through X-ray radiography. Each patient underwent a session of three applications of autologous platelet gel treatment with an interval of 3 to 4 weeks. Platelet gel was percutaneously injected into the nonunion site. Before each platelet gel injection, the patient was treated with a session of extracorporeal shock wave therapy. During the follow-up period, the X-ray radiographic and clinical examinations showed gradual formation of callus and improvement of clinical outcomes in almost all patients. The X-ray radiographic examination showed that complete union was observed in 12 patients and partial union was observed in 2 patients at about 4 to 6 weeks, while there was no treatment effect in 1 patient. The authors concluded that the use of autologous platelet gel combined with extracorporeal shock wave therapy is an efficient and promising method for reconstructing nonunion.

In a study presented by Chiang et al. [15], the authors showed a positive treatment effect of using bone graft mixed with autologous platelet gel on reconstructing atrophic nonunions. Twelve patients suffering from atrophic nonunions at the femur or tibia were enrolled and they had an average of 4.3 previous procedures including at least 1 (with an average of 1.5) autografting procedure for treating the nonunion, but all procedures had failed. The nonunions of patients had an average age of 24.6 months at the time of the enrollment. For each patient, an autologous bone graft or a bone graft complex (autograft with allograft) mixed with autologous platelet gel, created by spraying platelet gel onto the bone graft, was cut into a specific shape to fill the bone defect and to bridge both ends of the defect during the surgery. The residual platelet gel was sprayed onto both ends of the defect. The follow-up examinations showed that the nonunions of all patients had successfully healed, confirmed using X-ray radiography. Eleven patients successfully healed at an average of 19.7 weeks, while one patient successfully healed at 21 weeks. For each patient, the bone mineral density at the nonunion site increased continuously throughout 12 months and was higher than that at the contiguous site of the same limb and that at the matched site of the contralateral limb. Most of the functional outcomes including physical functioning, social functioning, mental health, body pain, and vitality were significantly improved. This study is among the first to show a promising positive treatment effect of using bone graft mixed with autologous platelet gel on reconstructing atrophic nonunions.

In a case study presented by Smrke et al. [50], the authors showed a positive treatment effect of using autologous bone graft mixed with allogeneic platelet gel on reconstructing a nonunion for a diabetic patient suffering from nonunion after a comminuted fracture at the tibia. The fracture was diagnosed as a delayed union by the authors since it showed no signs of healing 6 months after the trauma event and the following treatments. In the literature, it has been reported that the expression of growth factors and the potential of platelet-rich plasma for healing in autologous platelets of diabetic patients might be decreased [50,63,64]. Hence, in this study, instead of using autologous platelets, the authors chose to use allogeneic platelets to produce platelet gel for the treatment. The autologous bone graft was fragmented using a manual grinder and was mixed with allogeneic platelet gel, and then it was molded to fill the bone defect of the patient. The follow-up examination at 12 months after the surgery showed that the bone defect solidly bridged, while the affected bone restored the normal structure, confirmed using X-ray radiography and computed tomography. The patient was able to be fully weight bearing, and both legs of the patient were of the same length. No side effects were observed.

In a study presented by Mariconda et al. [51], the authors showed that there was no significant effect of using autologous platelet gel to shorten the healing time for atrophic nonunions at long bones treated using external fixation. In this study, autologous platelet gel was used alone and was not mixed with bone graft or other materials. Twenty patients with atrophic nonunions at long bones (femur, tibia, or forearm bones) were enrolled. The fractures of patients were diagnosed as nonunions since no callus formation could be observed using X-ray radiography at least 8 months after the injury. During the surgical treatment using external fixation, platelet gel was percutaneously injected at the bone defect under fluoroscopic guidance. The healing time of nonunions of 20 patients was compared with that of 20 matched controls with nonunions at long bones treated using external fixation without platelet gel injection. The results showed that the successful healing rate was 90% (18/20) in patients and 85% (17/20) in controls, respectively, while the healing time was 147 ± 63 days in patients and 153 ± 61 days in controls, respectively. There was no significant difference in the successful healing rate and healing time between both groups, failing to show the treatment effect of using autologous platelet gel on shortening the healing time for nonunions at long bones treated using external fixation. However, in this study, platelet gel was applied using percutaneous injection and was not applied to densely fill the bone defect during the surgery. It could be comprehended that platelet gel might not completely cover and densely fill the bone defect if it was applied using percutaneous injection; therefore, the treatment effect might be compromised.

In a case study presented by Dallari et al. [52], the authors showed a positive treatment effect of deep decortication in combination with bone graft mixed with autologous platelet gel and autologous bone marrow stromal cells on reconstructing a nonunion for a patient suffering from nonunion at the femoral diaphysis after a traumatic fracture. During the decortication surgery, the bone graft mixed with autologous platelet gel and autologous bone marrow stromal cells was applied to fill the bone defect at the nonunion site. Forty days after the surgery, the histological analysis showed signs of early bone healing response. Six months after the surgery, the patient was able to walk with full weight-bearing without pain. The X-ray radiography confirmed the successful healing of the nonunion. The authors concluded that deep decortication in combination with bone graft mixed with autologous platelet gel and autologous bone marrow stromal cells is effective in accelerating the healing of bone nonunions. However, since multiple treatments were used simultaneously, there was no evidence that the resulting positive treatment effect was due to the combined effect of these treatments or mainly due to a specific treatment. The results of this study could not confirm the treatment effect of platelet gel alone.

In a case study presented by Lusini et al. [53], the authors showed a positive treatment effect of using autologous bone graft mixed with autologous platelet gel on reconstructing a nonunion for a patient complaining of chest pain and exertional dyspnoea due to sternal dehiscence after coronary artery bypass grafting surgery. The X-ray radiographic and computerized tomographic examinations demonstrated incomplete closure of the sternal fracture, suggesting atrophic nonunion. During the revision surgery, the autologous bone graft mixed with autologous platelet gel was applied to fill the sternal gap. At the follow-up examination 4 months after surgery, the sensation of pain of the patient was significantly less, while the X-ray radiographic and computerized tomographic examinations demonstrated that the nonunion has been solidly bridged and completely healed.

## 4. Research on Using Platelet Gel to Reconstruct Bone Defects Associated with Surgery

In a study presented by Leonardi et al. [54], the authors showed a positive treatment effect of using allogenic bone graft mixed with autologous platelet gel on reconstructing bone defects for two patients treated for hip revision with a partial pelvis replacement ring. These two patients suffered from bone loss after the previous surgery; therefore, a revision surgery was required to restore the integrity of the bone. During the revision surgery, the allogenic bone graft mixed with autologous platelet gel was molded to fill the bone defect after positioning and fixing the ring. The authors reported that, after spraying platelet gel onto the bone graft, the platelet gel was absorbed by the bone graft in about 10 s. Thirty seconds later, the platelet gel started to retract, keeping the morselized bone graft tight but still plastic enough to be positioned under the ring. Then, after 2 to 3 min, the mixture of bone graft and platelet gel became solidified, keeping the bone graft firm. The authors believed that platelet gel can provide enhancement and help to keep the morselized bone graft firm, making it easier to fix the bone graft at the site of the bone defect and to densely fit the bone graft into the bone defect. The mean follow-up time was 27.2 months. For the two patients, the X-ray radiographic examinations confirmed the successful healing of the bone defect while the clinical examinations showed satisfactory outcomes.

In a brief report presented by Franchini et al. [55], the authors showed a positive treatment effect of using autologous platelet gel mixed with hydroxyapatite on reconstructing bone defects for 19 patients undergoing reconstructive bone surgeries. The indications for surgery include fibrous dysplasia, fracture, osteomyelitis, pseudoarthrosis, total hip arthroplasty with acetabular reconstruction for acetabular dysplasia, bilateral lower-extremity lengthening, bilateral hip arthroplasty with acetabular reconstruction for rheumatoid arthritis, or revision of hip prosthesis. In this study, autologous platelet gel was mixed with hydroxyapatite and then applied during the surgery, but the authors did not describe how they applied the mixture of platelet gel and hydroxyapatite during the surgery. The follow-up examinations using X-ray radiography after a median follow-up time of 12.9 months showed improved osteoblastic reaction and restoration of bone structure in all patients, while progressive incorporation of hydroxyapatite into the surrounding bone could be observed. The clinical outcomes of all patients were also satisfactory without complications. The authors concluded that the results of this study demonstrated the osteoinductive effectiveness of autologous platelet gel to stimulate bone regeneration following reconstructive bone surgery.

In a similar study presented by Franchini et al. [56], the authors showed a positive treatment effect of using allogenic bone graft mixed with autologous platelet gel on reconstructing bone defects for 25 patients who underwent reconstructive bone surgeries. The patients enrolled in this study had severe bone disorders (osteomyelitis, pseudoarthrosis, or bone defects due to unreported reasons) and had undergone several previous surgeries, but all surgeries failed to solve their problems. In this study, allogenic bone tissues from a tissue bank were fragmented using a mechanical miller, and the resulting bone chips were mixed with autologous platelet gel. The mixture of bone graft and platelet gel was applied to fill the bone defect during the surgery. The mean follow-up time was 16.9 months. The results of the post-operative follow-up showed satisfactory outcomes for all patients based on the biological, X-ray radiographic, and clinical examinations, although a patient underwent an amputation of the leg 8 months after the surgery due to the primary disease of that patient. The authors concluded that allogenic bone graft mixed with autologous platelet gel is efficient in the regenerative treatment of complex bone disorders.

In a study presented by Savarino et al. [57], the authors showed a positive treatment effect of using allogenic bone chips mixed with autologous platelet gel on reconstructing bone defects for patients undergoing high tibial osteotomy for treating genu varus due to osteoarthritis. Ten patients were enrolled and divided equally into a patient group and a control group. In the patient group, the mixture of lyophilized allogenic bone chips and autologous platelet gel was applied to fill the bone defect during the surgery. In the control group, lyophilized allogenic bone chips were applied to fill the bone defect alone without using platelet gel. The follow-up examinations 6 weeks after the surgery showed that there was no difference in the clinical and functional outcomes between both groups. However, the histological, histomorphometric, and microradiographic analyses showed signs of accelerated healing and regeneration (as evidenced by new bone and vessel formation) in the group with the use of platelet gel compared to the control group, although X-ray diffraction analysis showed that there was no difference in the microstructure between both groups. Based on the results of this study, the authors concluded that the mixture of allogenic bone chips and autologous platelet gel can promote osteogenesis and accelerate bone healing and regeneration.

In a similar study presented by Dallari et al. [58], the authors showed a positive treatment effect of using allogenic bone chips mixed with autologous platelet gel or allogenic bone chips mixed with autologous platelet gel and bone marrow stromal cells on reconstructing bone defects for patients undergoing unilateral opening-wedge high tibial osteotomy for treating genu varum. Twenty-eight patients were enrolled and divided into three groups. In group A (9 patients), the mixture of lyophilized bone chips and platelet gel was applied to fill the bone defect during the surgery. In group B (10 patients), the mixture of lyophilized bone chips, platelet gel, and bone marrow stromal cells was applied. In the control group (9 patients), lyophilized bone chips alone were applied. The histological and histomorphometric analyses 6 weeks after the surgery showed signs of accelerated healing and regeneration (as evidenced by new bone and vessel formation) in the groups A and B compared to the control group. The X-ray radiographic examinations 1 year after the surgery showed better osteointegration in the groups A and B compared to the control group. However, there was no difference in the clinical and functional outcomes among the three groups. The authors concluded that adding platelet gel or platelet gel combined with bone marrow stromal cells to lyophilized bone chips can enhance the osteogenetic potential of lyophilized bone chips for treating of massive bone loss, while complete osseointegration was not achieved in some patients with the use of lyophilized bone chips alone.

In a study presented by Feiz-Erfan et al. [59], the authors showed that there was no significant effect of using autologous platelet gel alone to enhance earlier fusion and increase fusion rate for patients undergoing anterior cervical fusion with allograft and internal fixation. Eighty-one disc levels were treated surgically in the 50 enrolled patients, in which 29 patients had degenerative cervical disc disease and 21 patients had herniated cervical disc disease. Forty-two disc levels were assigned to be treated using allograft mixed with autologous platelet gel, and thirty-nine disc levels were assigned to be treated using allograft without autologous platelet gel. The X-ray radiographic examinations 1 year after surgery showed that the overall fusion rate was 84% (68 of 81), and there was no significant difference in the fusion rate between patients with and without receiving platelet gel. For all patients, the outcomes of the pain and disability examinations showed significant improvements without significant differences. The authors concluded that the use of autologous platelet gel alone had no significant effect in enhancing earlier fusion and increasing fusion rate for patients undergoing anterior cervical fusion with allograft and internal fixation.

In a study presented by Sabbagh et al. [60], the authors showed a positive treatment effect of using autologous platelet gel on promoting the healing of phalangeal fractures after fixation by Kirschner wires for 20 patients. In this study, platelet gel was used alone and was not mixed with bone graft or other materials. The X-ray radiographic and clinical examinations showed signs of healing at 12 weeks for 70% of patients, and 16 weeks for another 30% patients. The authors concluded that autologous platelet gel was useful to promote healing and decrease recovery time in phalangeal fractures after fixation by Kirschner wires.

## 5. Research on Using Platelet Gel to Reconstruct Bone Defects in Patients with Bone Tumors

In a study presented by Loquercio et al. [61], the authors showed a positive treatment effect of using β-tricalcium phosphate, commercialized bone graft material and autologous platelet gel on reconstructing bone defects for 16 patients with giant cell tumors treated with curettage surgery. These three materials were applied simultaneously to fill the bone defect during curettage surgery. For all patients, the X-ray radiographic and computerized tomographic examinations showed complete healing at an average of four months, while the functional examinations showed satisfactory outcomes. Four patients showed recurrence during the follow-up period. The authors concluded that the use of autologous platelet gel as a supplementation can significantly reduce the time required for bone healing after curettage surgery for giant cell tumor and can achieve satisfactory functional outcomes.

In a study presented by Mostafa et al. [62], the authors showed a positive treatment effect of using autologous platelet gel mixed with hydroxy appetite/beta tri-calcium phosphate bone substitutes on reconstructing bone defects after extended curettage surgery for 20 patients with benign bony cystic lesions. The mean follow-up time was 18 months. The X-ray radiographic and clinical examinations showed satisfactory outcomes in all patients, although two patients showed recurrence. The authors concluded that autologous platelet gel mixed with hydroxy appetite/beta tri-calcium phosphate bone substitutes can be useful to reconstruct bone defects.

## 6. Discussion

To the best of the author’s knowledge, this is the first review paper that intends to conclude a current consensus on the treatment effect of platelet gel on reconstructing bone defects and nonunions. One of the hopes of the present review paper is to provide a valuable reference for clinicians, so that clinicians can decide whether they should apply platelet gel to reconstruct bone defects and nonunions in clinical practice or not. In addition, the author hopes the present review paper can motivate clinicians and researchers to conduct more clinical trials and clinical and basic studies to further justify the treatment effect of platelet gel on reconstructing bone defects and nonunions. In the present review paper, only 15 relevant studies were found using the designated keywords. This means that, to date, there are still few clinical studies intending to investigate the treatment effect of platelet gel on reconstructing bone defects and nonunions. More future clinical studies are encouraged to further investigate the effectiveness of this promising treatment method for reconstructing bone defects and nonunions.

Most of the studies reviewed in the present paper applied the combination of platelet gel and bone graft (autologous or allogenic) to reconstruct bone defects and nonunions. Of these studies, most of the results were positive, suggesting that the combination of platelet gel and bone graft could effectively enhance bone healing and regeneration and could be a promising and useful treatment strategy for reconstructing bone defects and nonunions. In this treatment strategy in which platelet gel and bone graft are mixed together, platelet gel can be regarded as a biomaterial used to tremendously enhance the treatment effect of bone graft. Numerous types and large amount of growth factors contained in platelet gel can largely enhance the healing and regeneration potential of the bone graft. In addition, the gelatinous and viscous platelet gel can help to strengthen the structure of the bone graft by keeping the bone graft firm, making it easier to fix the bone graft at the site of the bone defect and to densely fit the bone graft into the bone defect.

Only two studies to date tried to apply platelet gel alone to reconstruct bone defects and nonunions [51,60]. In these two studies, platelet gel was used alone, no other materials (such as bone graft) were mixed with platelet gel and no other treatment methods (such as extracorporeal shock wave therapy) were used in combination with platelet gel. The results of these two studies were conflicting. In theory, the application of platelet gel alone could promote the reconstruction bone defects and nonunions, since platelet gel contains numerous types and large amount of growth factors as well as provides an appropriate environment for cells and tissues to grow, which are essential for bone healing and regeneration. However, it should be noted that the treatment effect can be crucially influenced by the quality and characteristics of platelet gel, which can be largely determined by the method of preparation of platelet gel. Hence, an optimal protocol to prepare platelet gel is very important. In the future, more clinical studies must be conducted to understand whether the application of platelet gel alone is useful to reconstruct bone defects and nonunions or not.

Only one study to date applied autologous platelet gel combined with extracorporeal shock wave therapy to reconstruct nonunions [49]. The results of this study were positive and promising. In this study, each patient underwent a session of three applications of autologous platelet gel treatment with an interval of 3 to 4 weeks. Platelet gel was percutaneously injected into the nonunion site. Before each application of autologous platelet gel injection, the patient was treated with a session of extracorporeal shock wave therapy. The authors of this study hypothesized that extracorporeal shock wave therapy can induce microfractures and create a larger contact area on the bone surface. Hence, there would be a large contact area for platelet gel to lie on the bone while at the same time the growth factors released by platelet gel could penetrate into the bone through the microfractures, probably resulting in a significant enhancement of the healing potential of the growth factors. The promise of this reasonable hypothesized mechanism was reflected in the study results that, in almost all patients, the X-ray radiographic and clinical examinations showed formation of callus and improvement of clinical outcomes during the follow-up period. Complete union was observed in most of the patients. The promising findings of this study suggest that the use of autologous platelet gel combined with extracorporeal shock wave therapy could be a useful treatment strategy to reconstruct bone defects and nonunions. More future clinical studies are encouraged to further justify the effectiveness of this treatment strategy.

Basically, in the studies reviewed in the present paper, there are two methods to apply platelet gel treatment. In most of the studies, platelet gel was applied by densely fitting it into the bone defect during surgery. In other two studies, platelet gel was percutaneously injected into the site of bone defect [49,51]. It can be imagined that different application methods could lead to different treatment effects. If platelet gel is applied during surgery, it can densely fill and cover the bone defect and bridge the ends of the defect and can be fixed at the site of the defect; it can be imagined that the treatment effect could be maximized with this application method. However, this application method is invasive and must be applied during a surgery. On the other hand, the percutaneous injection of platelet gel can be more easily performed since it is minimally invasive in nature. However, it does not possess the above-mentioned advantages of surgical application of platelet gel, therefore how effective its treatment effect is remains unclear. In the future, more clinical studies are needed to further understand and compare the treatment effects of surgical application and percutaneous injection of platelet gel.

Most of the studies reviewed in the present paper lacked a matched control group, and it was a common limitation shared by these studies. However, the design of these studies was to investigate the treatment effect on recalcitrant atrophic nonunions that had been treated using previous clinical interventions but showed no signs of healing for a long period of time, and the results showed that most of the nonunions successfully healed after treating using platelet gel. Hence, the treatment effect could still be justified with such a study design even though there was no matched control group. Nevertheless, more clinical studies with matched control groups are still necessary to further justify the effectiveness of platelet gel to reconstruct bone defects and nonunions.

Since firstly introduced in the early 1990s, the application of platelet gel has been successful in numerous clinical fields including oral and maxillofacial surgery. In fact, reconstructing bone defects in oral and maxillofacial surgery is one of the fields in which platelet gel was firstly applied. Some promising results have been reported [65,66,67,68]. Since the focus of the present paper is on the field of orthopedics, the studies investigating the treatment effect of platelet gel on reconstructing bone defects in oral and maxillofacial surgery were not reviewed in the present paper.

## 7. Conclusions

In conclusion, based on the review of the present paper, the application of platelet gel is a promising and useful treatment method for reconstructing bone defects and nonunions. The information provided in the present paper not only can be useful for clinicians to make treatment decision in clinical practice, but also can motivate researchers to further investigate the treatment effect and underlying mechanisms of platelet gel on reconstructing bone defects and nonunions. The present review found that most of the studies applied the combination of platelet gel and bone graft to reconstruct bone defects and nonunions, and most of the results were positive, suggesting that this treatment strategy could promote successful reconstruction of bone defects and nonunions. Only two studies tried to apply platelet gel alone to reconstruct bone defects and nonunions; therefore, a convincing conclusion could not be made yet; more future clinical studies must be conducted to understand the treatment effect of platelet gel alone on reconstructing bone defects and nonunions. Only one study applied platelet gel combined with extracorporeal shock wave therapy to reconstruct nonunions, and the results were positive; the hypothetical mechanism of this treatment strategy is reasonable and sound, and more future clinical studies are encouraged to further justify the effectiveness of this promising treatment strategy. Platelet gel can be applied by densely fitting it into the bone defect during surgery, or can be applied through percutaneous injection; it can be imagined that these two application methods are fundamentally different and could lead to different treatment effects, and more future clinical studies are needed to further understand the difference in the treatment effect between these two application methods. Overall, in the present review paper, only 15 relevant studies were found using the designated keywords. This means that, to date, there are still few clinical studies intending to investigate the treatment effect of platelet gel on reconstructing bone defects and nonunions. More future clinical studies are encouraged to further investigate the effectiveness of this promising treatment method for reconstructing bone defects and nonunions.

## Figures and Tables

**Figure 1 ijms-23-11377-f001:**
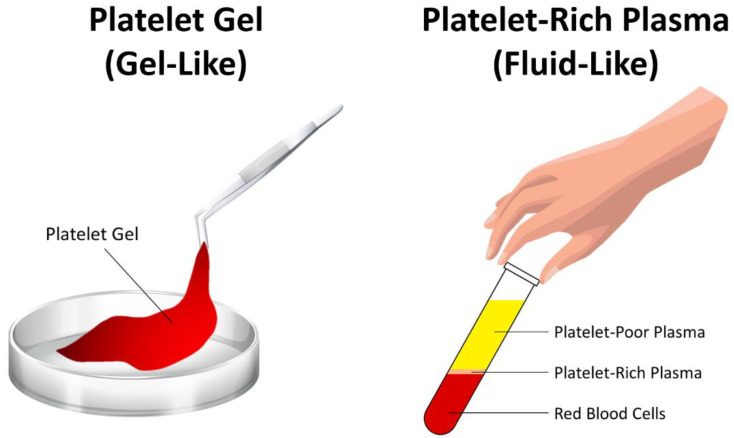
Illustration of gel-like platelet gel and fluid-like platelet-rich plasma.

**Figure 2 ijms-23-11377-f002:**
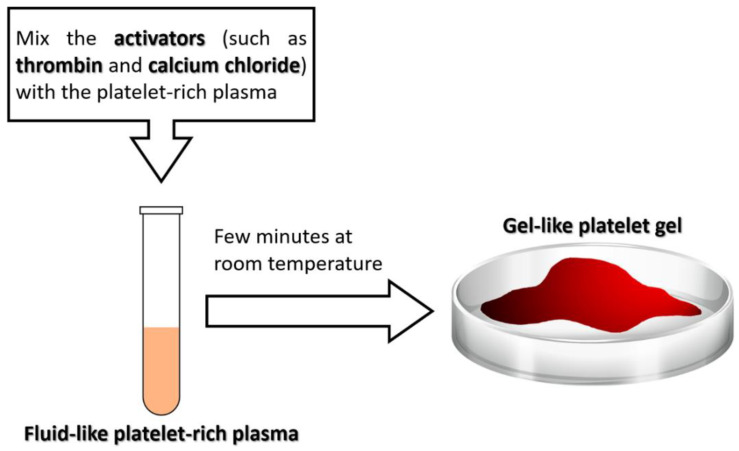
Different from fluid-like platelet-rich plasma, platelet gel is a gel-like material rich in platelets as its name suggests and is rapidly formed when platelet-rich plasma is mixed with activators such as thrombin and calcium chloride.

**Figure 3 ijms-23-11377-f003:**
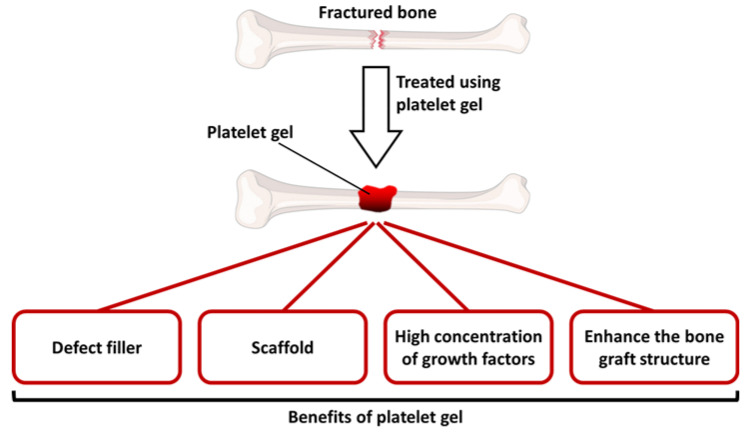
Illustration of advantages of platelet gel on reconstructing bone defects and nonunions.

**Table 1 ijms-23-11377-t001:** Summary of advantages of platelet gel on reconstructing bone defects and nonunions.

Unlike platelet-rich plasma which is fluid-like, platelet gel is gel-like and therefore can densely fill the defect and can be fixed at the site of the defect. Platelet gel contains numerous types and a large amount of growth factors that are essential for promoting healing and regeneration. Platelet gel can serve as a natural scaffold, providing an appropriate environment for cells and tissues to grow. Platelet gel can enhance the structure of the graft and keep the graft firm when mixing with the bone graft, while the mixture of graft and platelet gel can be molded to the desired shape to fill the defect and bridge both ends of the defect.

**Table 2 ijms-23-11377-t002:** Summary of the patient number, treatment method, and effect of each reviewed article. Please note that the treatment methods described in this table only include those directly relevant to the application of platelet gel. The surgical and other clinical interventions involved in each article (if there were any) that seemed not directly relevant to the application of platelet gel are not described in this table.

References	Patient Number	Treatment Method	Treatment Effect
Rughetti et al. [49]	15 patients	Autologous platelet gel combined with extracorporeal shock wave therapy	Complete union in 12 patients and partial union in 2 patients with improved clinical outcomes, while no effect in 1 patient
Chiang et al. [15]	12 patients	Bone graft mixed with autologous platelet gel	Nonunions of all patients successfully healed with improved clinical outcomes
Smrke et al. [50]	1 patient (Case study)	Autologous bone graft mixed with allogeneic platelet gel	Nonunion of the patient successfully healed with improved clinical outcomes
Mariconda et al. [51]	40 patients, with 20 in the experimental group and 20 in the control group	Autologous platelet gel alone	The successful healing rate was 90% (18/20) in patients in the experimental group and 85% (17/20) in controls, respectively; there was no significant difference in the successful healing rate between both groups
Dallari et al. [52]	1 patient (Case study)	Bone graft mixed with autologous platelet gel and autologous bone marrow stromal cells	Nonunion of the patient successfully healed with improved clinical outcomes
Lusini et al. [53]	1 patient (Case study)	Autologous bone graft mixed with autologous platelet gel	Nonunion of the patient successfully healed with improved clinical outcomes
Leonardi et al. [54]	2 patients	Allogenic bone graft mixed with autologous platelet gel	Bone defects of all patients successfully healed with improved clinical outcomes
Franchini et al. [55]	19 patients	Autologous platelet gel mixed with hydroxyapatite	Bone defects of all patients successfully healed with improved clinical outcomes
Franchini et al. [56]	25 patients	Allogenic bone graft mixed with autologous platelet gel	Bone defects of all patients successfully healed with improved clinical outcomes
Savarino et al. [57]	10 patients, with 5 in the experimental group and 5 in the control group	Allogenic bone chips mixed with autologous platelet gel	Signs of accelerated healing and regeneration were observed in patients in the experimental group, while there was no difference in the clinical and functional outcomes between both groups
Dallari et al. [58]	28 patients, with 19 in the experimental group and 9 in the control group	Allogenic bone chips mixed with autologous platelet gel or autologous platelet gel combined with bone marrow stromal cells	Signs of accelerated healing and regeneration were observed in patients in the experimental group, while there was no difference in the clinical and functional outcomes between both groups
Feiz-Erfan et al. [59]	81 disc levels in 50 patients were treated, with 42 disc levels in the experimental group and 39 disc levels in the control group	Allograft mixed with autologous platelet gel	The overall fusion rate was 84% (68 of 81), and there was no significant difference in the fusion rate between both groups
Sabbagh et al. [60]	20 patients	Autologous platelet gel alone	Bone defects of all patients successfully healed with improved clinical outcomes
Loquercio et al. [61]	16 patients	β-tricalcium phosphate, commercialized bone graft material and autologous platelet gel	Bone defects of all patients successfully healed with improved clinical outcomes
Mostafa et al. [62]	20 patients	Autologous platelet gel mixed with hydroxy appetite/beta tri-calcium phosphate bone substitutes	Bone defects of all patients successfully healed with improved clinical outcomes

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
