# Peer review of "Treatment Effect of Platelet Gel on Reconstructing Bone Defects and Nonunions: A Review of In Vivo Human Studies"

_ijms, 2022, doi:10.3390/ijms231911377_

Round 1

Reviewer 1 Report

The paper is well organized and well written.

I suggest improving the graphics of Figure 2

Author Response

Dear Reviewer,

Please find the attached file for the response letter for you.

Thank you very much for your time and efforts to carefully and thoroughly review the manuscript, as well as for your important comments and constructive suggestions that help to improve the manuscript substantially. I am very grateful and highly appreciate. Thank you very much!

Best wishes sincerely,

Che-Yu Lin

Reviewer 2 Report

The authors reviewed research articles dedicated to the Treatment Effect of Platelet Gel on Reconstructing Bone De-2 fects and Nonunions.

 There are several problems to be addressed:

  There are several publications related to this subject. Include few references from “International Journal of Molecular Sciences”.

 Table 1 is not informative. The authors should add some more details regarding the amount of positive and negative reactions of patients after gel introducing.

 I also suggest adding the conclusion section. The authors should outline the main conclusion of the paper.

I recommend to add more illustrative materials and tables for better understanding of the paper.

Author Response

(The authors gave the same response as above.)

Reviewer 3 Report

- Figures 1 and 2 are too minimal and should be combined. Also, Figure 2 is merely showing some text boxes. Authors must add some schematics or real pictures of the reagents and the product.

- Figure 3: this seems to be better presented in the form of a table. Again, presenting all textual data as a figure is not standard. I recommend converting this to a table, and then adding much more details to the list. Both advantages and disadvantages of each system must be presented here.

- Table 1 is very informative. There are, however, a number of improvements that must be made: A column could be added to delineate the species/animal model that was used in each study. Were there any in vitro studies also listed in this Table? Also, the last column, the Treatment Effect, is quite vague by just saying 'positive'. More specific info should be presented for each study, as the main findings and impact of the treatment.

- Authors do not elaborate the in vitro studies that have been conducted using this gel system to model/examine bone regeneration. This is worth a separate subsection, and probably a figure.

Author Response

(The authors gave the same response as above.)

Round 2

Reviewer 2 Report

Manuscript can be accepted in present form in IJMS

Reviewer 3 Report

The revised manuscript is appropriate for publication.